# Barriers to the Implementation of Radio Frequency Identification (RFID) for Sustainable Building in a Developing Economy

Ahmed Farouk Kineber [1,2,*], Ayodeji Emmanuel Oke [3,4,5], Mohamed Elseknidy [6], Mohamed Magdy Hamed [7] and Fakunle Samuel Kayode [3]

1   Department of Civil Engineering, College of Engineering in Al-Kharj, Prince Sattam Bin Abdulaziz University, Al-Kharj 11942, Saudi Arabia
2   Department of Civil Engineering, Canadian International College (CIC), 6th October City, Zayed Campus, Giza 12577, Egypt
3   Department of Quantity Surveying, Federal University of Technology, Akure 340110, Nigeria
4   CIDB Centre of Excellence, Faculty of Engineering and Built Environment, University of Johannesburg, Johannesburg 2092, South Africa
5   School of Social Sciences, Universiti Sains Malaysia, Penang 11800, Malaysia
6   Department of Chemical and Environmental Engineering, Faculty of Engineering, University of Putra Malaysia, UPM, Serdang 43400, Malaysia
7   Building and Construction Engineering Department, College of Engineering and Technology, Arab Academy for Science, Technology and Maritime Transport (AASTMT), B 2401 Smart Village, Giza 12577, Egypt
*   Correspondence: a.kineber@psau.edu.sa

**Abstract:** Sustainability principles need to be applied at all the stages of the decision-making process concerning the building of urban housing to realize maximum benefits without compromising the project's function. This paper aims to investigate the applicability of radio frequency identification (RFID) and identify the barriers that impede its successful adoption in building projects to achieve sustainability in building. The literature was reviewed, and data were derived by complementing the quantitative technique. A questionnaire was employed to gather data from 107 stakeholders in the building industry in Nigeria. The data were scrutinized using the exploratory factor analysis (EFA) technique. The partial least square structural equation modeling (PLS-SEM) was also applied to create a model for embracing RFID tools for sustainable building. The results of EFA revealed that the RFID barriers could be classified into significant constructs: infrastructure, immaturity, privacy, and security. The PLS-SEM model revealed that infrastructure was the most significant barrier to RFID implementation in the building industry. Thus, this study's findings could aid decision makers in facilitating sustainability approaches in buildings projects through RFID implementation. These results would further lay the basis for objectively measuring and valuing the diverse barriers impacting RFID implementation.

**Keywords:** radio frequency identification; building projects; project performance; sustainability; partial least square; structural equation modeling

## 1. Introduction

The building business is complex and dynamic. Buildings projects accounted for 30% of the total greenhouse gas (GHG) releases in both developed and emerging economies. Building projects also accounted for over 40% of global power generation [1]. Estimates revealed that 40% of Europe's and the USA's energy and resources are devoted to building activities [2]. Sustainable building activities in emerging economies are somewhat worrisome [3]. Although these countries are fast-growing, there are fewer efforts by the building sector concerning guaranteeing the needed standard of living [4]. Beach et al. [5] contended that the building industry is an exceptionally decentralized, data-intensive and

project-based market. It needs substantial data processing and sharing throughout its lifecycle properties contingent on various companies and different experts in building.

The design approach, overseeing and upgrading a structure, comprised traditional disciplines, including electrical and mechanical engineering, structure and architecture. It likewise involved many modern disciplines, including environmental science and waste management. Most of these works have high technical requirements. The recently evolving wireless tools, including radio frequency identification (RFID), have transformed from anonymity to conventional applications that aid the quick handling of produced materials and goods [6]. RFID has existed for many decades.

RFID allows detection from a distance and is different from barcode tools. RFID does not require a direct line of sight. Conversely, it has only recently merged with increased capabilities and low costs, requiring industries to thoroughly analyze what RFID can offer. RFID applications are broad and comprise the distribution and manufacturing of material goods, including transmission and mobile fabrication [7] and the packaging process in pharmaceutical companies [8]. RFID utilizes electromagnetic fields to mechanically detect and trace the attached tags to the items [9]. The RFID system comprises a small radio transponder, a transmitter and a radio receiver. After activation by an electromagnetic interrogation pulsation from an immediate RFID reader maneuver, the digital data will be transmitted back to the reader by tags, typically the detecting register code [10]. This code can be employed to trace registered goods.

Kibert [11] argued that sustainability necessitates a sustainable ecosystem with ecological and sustainable energy use. Conversely, it was expounded as an initial procedure before and after building firms completed the task [12]. The building sector needs to be modernized by introducing novel, viable and effective building approaches [13]. The potential offered by RFID cuts across several activities. The success of a project is pursued by boosting sustainable components and activities at the beginning of the building task. In recent building projects, RFID utilized the project's planning and design building scheme [14]. Likewise, RFID can be utilized for asset monitoring in and out of building sites [15]. After activation by electromagnetic interrogator pulsation from an immediate RFID reader gear, the digital data will be transmitted by the tags, typically a detecting mechanism for registered code, back to the tracker [10]. The code can be utilized to trace the registered goods. Thus, there are two types of RFID tags: (i) reflexive tags that are driven by the RFID detector's energy from interrogating radio currents, and (ii) active tags that are driven by battery and, consequently, can be detected over a longer distance away from the RFID tracker, up to hundreds of meters [16]. Although the literature has digested the benefits of RFID, less effort has been put forward to control the benefits of RFID within the building sector in emerging economies.

RFID was presented as an innovative technology not extensively endorsed in the building business [15]. Nonetheless, not much has been realized concerning RFID applications in actual building activities despite the need for this tool. The possible reasons are numerous. Managers of building projects are unaware of RFID's potential benefits in the building. Diverse fiscal, technical or moral issues might likewise avert it from being extensively implemented in this mixed industry [17]. Therefore, this research can further add to the current works on RFID as an emerging research field by exploring the challenges and barriers concerning its adoption in Nigeria's building sector. This paper will highlight RFID research as an emerging field by analyzing issues associated with its implementation by the building industry in Nigeria. Despite the practical challenges within the building industry, there is a dire need for measuring the perception of critical players in the building sector. Therefore, this paper aims to evaluate the RFID implementation activities in Nigeria's building sector in Lagos as a case study, measure the relevant fields and define the fundamental issues that avert RFID applications in the building sector. This study's findings will aid policy makers in doing well concerning sustainable building plans by enforcing RFID.

Additionally, the results generated from this study might influence how building tasks are carried out within the Nigerian framework and in similar emerging nations where similar building schemes are carried out [18]. Due to the global–local framework approach used by this research, the significance of this study to the world was emphasized. Furthermore, the importance of this research might be illustrated by revealing its significance in local and international frameworks [19]. Consequently, that intelligibility is accomplished by aiming at emerging nations. Nigeria was employed as the local framework in this paper, i.e., instituting the significance. Therefore, a practical building scheme in Nigeria and comparable emerging economies might be realized by eliminating unnecessary expenditure and enhancing productivity using RFID [18]. The outline of this paper comprises the state of the art concerning building success, trailed by the research approach employed. Subsequently, the results of this study are discussed in comparison with the current literature. The significant findings and recommendations are summarized in the conclusion, directing future research.

## 2. Sustainable Building and RFID

### 2.1. RFID Application for Sustainable Building

There has been a lot of emphasis on sustainability in the literature [20]. Business transformation sustainability, goals and introduction of designs are complex undertakings [21]. Therefore, there is a need for environmental, social and economic sustainability [22]. The introduction of sustainability in the building sector has examined the accuracy of methods to encompass this idea into current operating circumstances [23]. Firms' search for environmental protection and defining social responsibility ethics can boost the extensive adoption of RFID as a critical strategic stage. The introduction of RFIDs can boost the establishment of sustainable building partners. Applying the RFID method by building associates allows storage and access to stored data instantaneously [24,25].

The history of RFID is long and started with its application during World War II and has developed to its modern applications [9]. The basic RFID design comprises a reader, tag and middleware to conduct sophisticated data analysis. This makes RFID practical for use in numerous applications with valuable results [26,27]. However, many problems emanate from using inert tags because of the nature of the system, significantly, and the degree of inconsistent readings within the untreated data [28]. The actual commencement of RFID would not have come until after the merging of two technologies was realized roughly around the periods of the two World Wars. The earliest technology was uninterrupted wave radio production, which was established in 1906. Subsequently, the radar system is supposed to have been established in 1922 and was extensively used during World War II [29]. The integration of these two technologies gave birth to RFID, first proposed academically in principle by Harry Stockman in 1948. During this period, RFID was utilized to differentiate between allied and enemy planes during the war. As noted by Stockman, the technology had not advanced to the point that fulfilling the RFID technological potential could be established [30].

However, Conway et al. [31] argued that research on RFID continued to be carried out by the military aeroplanes division and the academic community, who tried to establish foe and identification friend (IFF) equipment in the 1950s. This was not achieved until the 1960s, when article surveillance and a sensor Matic contained a security structure integrating RFID tags, which only kept an 'off and on' order to avert stealing in stores. In the 1970s, the RFID focused on animal and vehicle tracking and factory automation. This adoption of the RFID system finally gave birth to the first RFID assimilated model toll developed in 1978. Subsequently, it was utilized in different countries around the world. Later, in 1990, RFID was assimilated into societies' regular activities. These include the RFID key card utilization for improved security to allow a greater integrality level for protected sites [32].

Moreover, RFID enhances small and medium enterprises (SMEs) in their critical, cost-effective and sustainable business and creativity. Firms should attain proficiency in success-

ful business value improvements by generating an operational commercial model [33]. The total resources comprising effort and time are typically dedicated to a professional IT unit and apportioned to other critical departments in the company. The SMEs whose significant output is not IT-based should not be troubled with revolutionizing and safeguarding their information systems (IS). Preferably, the SMEs ought to emphasize their primary area to boost their companies' accomplishment and attractiveness.

Conversely, it should make SMEs more creative and explore modern approaches to doing viable business [34]. Adopting RFID can improve firms' output to realize targets while cutting unnecessary expenditures. RFID adoption can make SMEs more cost-effective and improve their productivity and business performance by focusing on their central transaction rather than nonfundamental businesses, including structural modernization and management.

### 2.2. How RFID Operates

The electronic product codes (EPC), RFID and the EPC international network were premeditated by the Auto-ID Centre to substitute the barcode technique. The Auto-ID Centre corporation was established in 1999 by five prominent research institutions, about one-hundred merchants, consumer products manufacturers and software firms [35]. Analogous to the barcode, the EPC is an exclusive digit that detects a particular item within the supply chain and is classified into digits that classify the product type and manufacturer. In contrast to the barcode, the EPC utilizes an extra set of numbers for a serial number to detect particular stuff. Where the barcodes only differentiate between goods, the EPC codes are exclusive to each unit and can deliver more comprehensive data [36]. The EPC has a version digit that describes the range of EPC between a collection of potential structures and can be utilized to define various types of available coding systems, including global trade identification number (GTIN), global returnable asset identifier (GRAI), global location number (GLN) and serial shipping container code (SSCC). The stored EPC in the RFID tag is higher than the barcode since the operator is not required to identify where an item is and does not require getting nearer to scan it [36].

Within a typical RFID system, RFID tags are glued to the items. The tags are motorized and drive out EPC data when identifying a sign from the label reader [37]. After retrieving EPC from the tag, it can be correlated to dynamic data, e.g., siting of origin and product movement laterally to the supply chain. The collected information by the RFID detector will be forwarded to a savant, middleware, which manages the information [38]. The managed data will be forwarded to backend databanks to be used by business schemes such as manufacturing design systems, supply chain management systems and inventory management systems [39]. For fast and effective distribution of EPC codes, the Auto-ID Centre created a network system—EPC global network—via the Internet Protocol (IP). The network enables parties within the supply chain to obtain up-to-the-minute information regarding transferring products and item data. The producer and trading associates in the same network develop and keep serialized data concerning every item.

The producer keeps and processes the data concerning the group of the produce, though the transaction partners host and manage the data concerning the movement of products via the supply chain [40]. As soon as any portion of the supply chain needs goods or goods transfer data, they forward a request for specific EPC data to the object name service (ONS). The international lookup service is provided to interpret an EPC into one or more Internet uniform reference locators (URLs). These URLs provide comprehensive data concerning the production and movement of the item, which is kept with the producer or business associates in a product mark language (PML) format [41]. The PML is built on the extensive mark-up language, XML [42].

### 2.3. Barriers to RFID Implementation for Sustainable Building

There is a dire need to moderate barriers to RFID adoption to guarantee quality control so that investors in the building industry can be accurately utilized and executed [43].

However, the major technological hurdle to adopting RFID technology in the building sector is the lack of an international standard format [44]. Suppose that each trading partner uses a distinct cost format for RFID implementation; that may be exorbitant. Nevertheless, business organizations support standardization even if particular trailers may be reluctant to share their economic improvement attained through adequate inventory tracing or supply chain proficiency. Another barrier is related to RFID tags and scanners.

The RFID labels produce waves which are grasped by water and some liquids, and the waves are correspondingly mirrored by metals, which can result in inaccurate readings [45]. In addition, other contractors can decline to obey the strict operation schedule due to the high cost connected to evolving it. The implementation by larger shops such as Tesco and Walmart could lower the tags' costs, a line of reasoning that can be applied to persuade other performers concerning the advantages of financing new technologies [46]. Notwithstanding the numerous advantages of RFID implementation, numerous companies are in the preliminary phases of exploring and data collection concerning the RFID application [47].

Studies from Sweden revealed that larger grocery retailers, including Coop, Axfood and ICA, have attempted a few experiments, though they are hesitant to adopt the technology's general uncertainty and the cost of investments [24]. Likewise, IKEA has made some trials. Organizations within other industries such as VOLVO Technology and SSAB Oxelösund have adopted RFID to a certain level for tracing purposes to recognize and track products [26]. RFID tools have not been implemented without issues with the tags or radio waves.

The literature also revealed that businesses implementing the RFID technology were required to conduct a structural tactical review of companies' business operations and interactions with distributors and suppliers [48]. Apart from the legal, organizational and technical hurdles of RFID, it will not surface without motivation for retailers and producers to implement the technology [48]. Table 1 presents various RFID barriers in the building business within the context of the existing literature.

**Table 1.** RFID barriers in the building industry.

| SN. | Barriers | Studies |
|---|---|---|
| 1 | High cost | [49,50] |
| 2 | Security | [51–53] |
| 3 | Lack of industry standard | [54–56] |
| 4 | Lack of maturity | [55,57] |
| 5 | Privacy | [58,59] |
| 6 | Low technical ability | [60] |
| 7 | Power availability | [59,60] |
| 8 | Inadequate training | [59,60] |
| 9 | Lack of knowledge about the technology | [53,54,61] |
| 10 | Time frame to develop the technology | [59,61] |
| 11 | Immaturity | [57,62] |
| 12 | Low investment | [49,50] |
| 13 | Commitment from other project participants | [53] |
| 14 | Maintenance | [60] |
| 15 | Lack of demand | [53] |
| 16 | Human issues | [60] |
| 17 | Unclear needs | [59,63] |
| 18 | Lack of communication in remote areas | [53] |
| 19 | Virus threats | [63] |
| 20 | Inefficient use of software | [57,62] |
| 21 | Low infrastructure | [55,57] |
| 22 | Complexity and difficulty in changing the software | [54] |

## 3. Research Design and Methods

This study aims to boost the successful sustainable building sector delivery in Nigeria by exploring and identifying the radio frequency identification (RFID) implementation barriers. Figure 1 depicts the phases of the study. The literature was explored to identify the RFID implementation barriers. Consequently, the questionnaire tool was developed to assess these obstacles. The answers obtained from the survey tool regarding the participants' perceptions of project delivery, particularly architects, quantity surveyors, engineers and builders, contractors, special contractors, operators, heavy-duty contractors, management professionals, staff, building managers, staff and operators of the building location are all stakeholders in the building business.

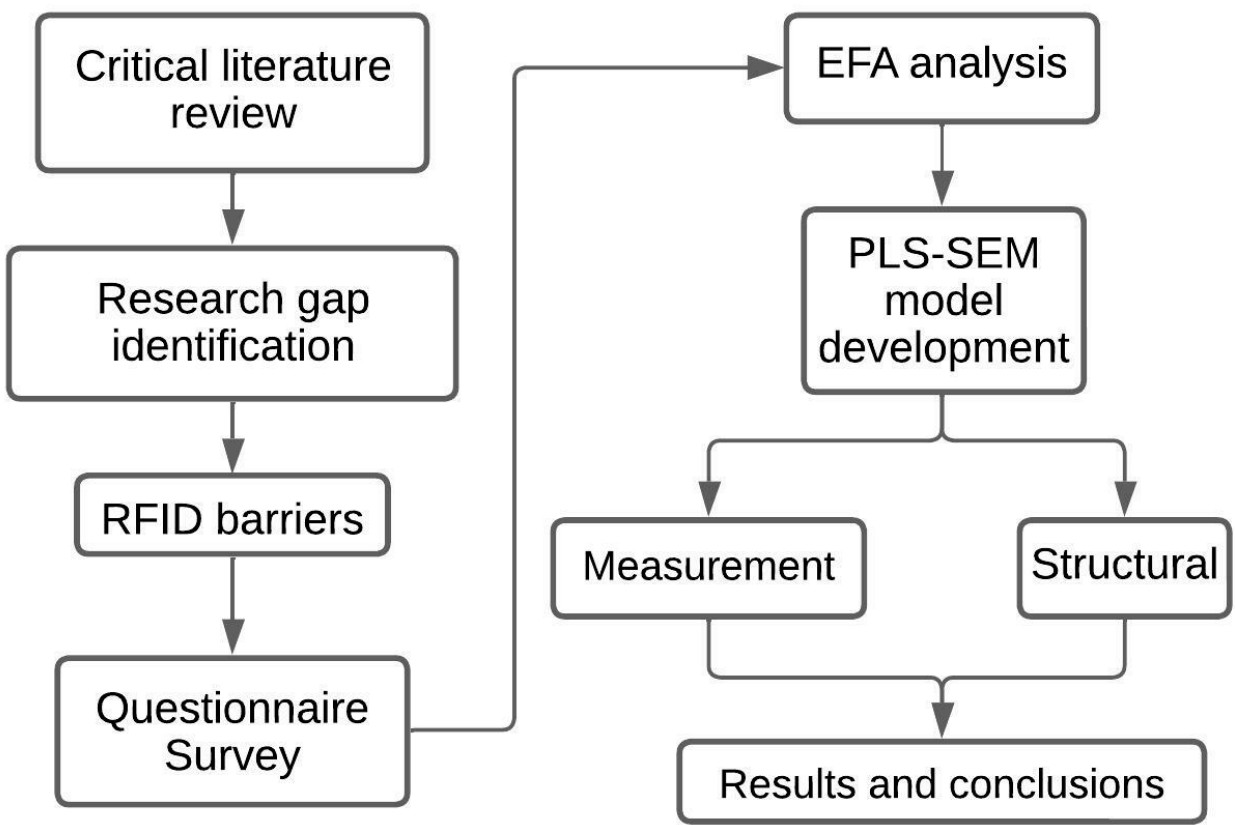

**Figure 1.** Research design.

### 3.1. Exploratory Factor Analysis (Analysis of Construct Validity)

Exploratory factor analysis (EFA) and confirmatory factor analysis (CFA) are the most widespread forms of factor analysis [64]. In this research, CFA was employed to analyze the fundamental features of variables in a specific hypothesis. Conversely, the EFA was utilized to understand the relationships between numerous components and reduce the variables into interpretable forms [65]. Principal component analysis (PCA) requires no initial premise in describing preliminary results within EFA [66]. Thompson [67] argued that PCA is the default form in different statistical packages and is extensively used in EFA. The Varimax rotation method is employed to Direct Oblimin or Promax, and the load distribution between parameters is curtailed [68]. The Varimax rotation is more appropriate in PCA. A typical unresolved theory curtails aspects [69]. The sum of variables would be perceived as an archetypal sample around the applicable ranges [70]. Consequently, twenty-three (23) considered parameters and the directed questionnaires to one-hundred and seven (107) participants were used to produce data for this paper and were considered appropriate for PCA [64].

### 3.2. Developing PLS-SEM Model

The partial least square and structural equation modeling (PLS-SEM) has attracted much interest from numerous disciplines, particularly in business and social sciences research [71]. Numerous types of research that focused on the PLS-SEM procedure have been presented in trending journals [72,73]. The modern SMART-PLS 3.2.7 software was used for data analysis and modeling of RFID instruments' significance via SEM. The PLS-SEM was initially endorsed for its powerful prediction principles compared to covariance-based SEM (CB-SEM); however, the variances among the binary methods are relatively low [74]. The statistical analysis conducted in this paper included measurement and structural assessment techniques.

### 3.2.1. Common Method Variance

The common method variance (CMV) can be described as a variance correspondence that can be attributed to concepts and the forms of measurement tools used [75]. Occasionally, personal data could be exaggerated or forestalled at the level of analyzed relationships and accordingly prompt challenges [76]. This could be critical for this study since all the analyzed data are personal, one-sided and resulting from a single source. Considering these issues is critical to detect changes in standard procedure. Harman (1976) described a formal one-factor test and was employed by Podsakoff and Organ [77]. A particular factor was obtained from the factor analysis, which exemplified a large percentage of the variance [76].

### 3.2.2. Measurement Model

The dimension (or measurement model) shows the relationship between variables and their underlying hidden structure [78]. The dimension model's discriminant and convergent validity are described in the subsequent sections.

#### Convergent Validity

Convergent validity (CV) demonstrates the extent of agreement between two or more indicators or tools of an analogous construct [79]. It is acknowledged as a subset of the construct validity. Concerning PLS, the CV of the estimated constructs can be defined using three (3) tests [80]: average variance extracted (AVE), Cronbach's alpha ($\alpha$) and composite reliability scores (Pc). Nunnally and Bernstein [81] recommended a Pc value of 0.7 as the higher limit of acceptable composite consistency. For any analysis, values above 0.7 and 0.6 for exploratory analysis are deemed acceptable [82]. The latest was the AVE, a standard measure employed to assess the constructs of CV in the dimension model. CV values higher than 0.50 are deemed acceptable [82].

#### Discriminant Validity

The discriminant validity (DV) recommends that the studied event is methodically special and designates that any dimension does not identify the peculiarity being evaluated in SEM [83]. Campbell and Fiske [84] contended that the correlation among pointers or tools being diverse from one another should be much higher for DV to be conducted.

#### Structural Assessment Model

The objective of this study was to explore and prioritize RFID implementation barriers using the SEM approach. The path coefficients between the estimated coefficients must be acknowledged to realize this. Therefore, the fundamental relationship or path relation was theorized between RFID £ tools and μ RFID barriers. Consequently, the fundamental relationship between *£*, *μ and* €1 rule within the structural model, which is identified as internal correlation, could be portrayed as a linear equation [85]:

$$\mu = \beta \pounds + \text{€}1 \tag{1}$$

where (β) is the path coefficient linking constructs of RFID adoption barriers, (€1) is the structural intensity residual variance likely to exist and β is the standardized weight of regression, similar to the β weight in the multiple regression model. The signs must be simultaneous with the model estimations and empirically weighty. Concerning CFA, a bootstrapping method found in the SMART-PLS 3.2.7 software package was used to compute the standard errors of the path coefficient. It was performed on 5000 subsamples based on the Henseler et al. [71] recommendation, which marks out the statistics for analyzing the hypothesis. Moreover, four (4) structural equations concerning constructs for RFID implementation barriers were created for the PLS model, indicating the internal correlations of concepts and Equation (1).

## 4. Results

### 4.1. The EFA of RFID Barriers

This research concentrated on RFID adoption barriers in Nigeria's building industry. The sampling procedure employed in this study has expediently facilitated data collection from the identified participants. The investigational methodology was adopted due to the size of the study population. EFA's research samples must be at least 45 to 61 study populations. Conversely, all the appropriate statistical tests are performed [86]. In this research, 122 participants were studied, and the required threshold of participants is 107. It constituted 87% of the return rate and was deemed acceptable for further analysis [86]. The participants' demographic features were obtained in the first section of the survey tool. Section two dealt with the RFID tools employing a 5-point Linkert Scale: Very High (5), High (4), Average (3), Low (3) and Very Low (1) (Supplementary Materials).

Numerous explicitly outlined factors for the factorability of correlation were used. The Kaiser–Meyer–Olkin Measure (KMO) can be assumed to estimate factor similarity, and it is extensively applied to estimate whether the fractional relationships between variables are the least conceivable [87]. Based on the KMO sampling adequacy estimator, the data return rate was acceptable for performing factor analysis. In the same vein, it is appropriate for Bartlett's sphericity test for correlation suitability between the very effective tools. The test revealed if the sampling approach or the data set is acceptable for factor analysis. The sampling suitability test was performed employing KMO = 0.716, which is suitable for factor analysis [87]. The findings revealed that the measured *p*-value had an estimated $Chi^2$ = 959.535. Thus, the analysis considered Bartlett's test significant ($p = 0.00$). It implied a significant relationship within the data matrix. It further revealed that all the listed variables' correlation matrix was strongly correlated at a 0.50 level. Therefore, the EFA's output was acceptable [88]. The total variance describes the RFID barriers' domains in the building industry. The PCA revealed the existence of four (4) components with eigenvalues above 1. These components described 55.16% of the total variance. Figure 2 shows a scree plot that depicts the eigenvalues on the *y*-axis and the variables on the *x*-axis. It further indicated a downward trend. The point where the slope of the curve is flattening off shows the total components that the model must produce.

It is noteworthy that among the significant reasons for performing factor analysis was to lessen the number of variables that explain a multifaceted construct of the RFID barriers within the component matrix concerning Nigeria's building industry. The rotated factor matrix concerning RFID adoption barriers in Nigeria's building industry is presented in Table 2. The model had four major components or barriers and was appropriate for signifying the relevance of RFID in Nigeria's building industry. Initially, while explaining the four major components, it is essential to highlight the four factors or variables. Each factor measures significantly onto only one of the collections. Thus, security, infrastructure, immaturity and privacy were employed to group the constructs into minor factors based on the literature after dropping 'privacy' due to the insignificant loading.

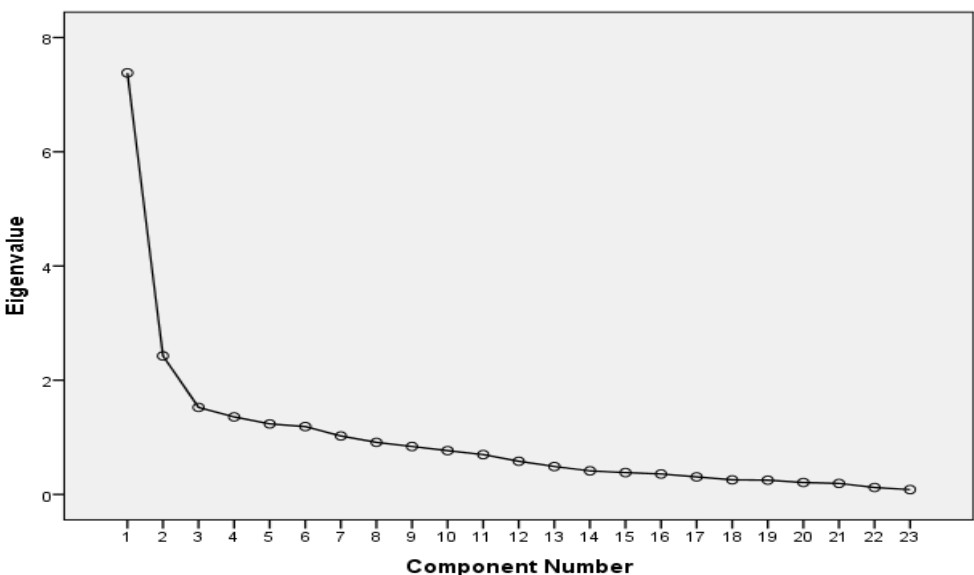

**Figure 2.** The scree plot of loading of barriers in implementing RFID technology in the building industry.

**Table 2.** Rotated component matrix on RFID barriers in the building industry.

| Rotated Component Matrix | | | | | |
|---|---|---|---|---|---|
| **Code** | **Name** | **Components** | | | |
| | | **1** | **2** | **3** | **4** |
| | Infrastructure | | | | |
| In1 | Lack of maturity | 0.671 | | | |
| In2 | Low infrastructure | 0.592 | | | |
| In3 | High cost | 0.581 | | | |
| In4 | Difficulty in changing the software | 0.559 | | | |
| In5 | Human issues | 0.557 | | | |
| In6 | Lack of industry standard | 0.533 | | | |
| In7 | Lack of communication in remote areas | 0.521 | | | |
| In8 | Maintenance | 0.520 | | | |
| | Immaturity | | | | |
| Im1 | Virus threats | | 0.808 | | |
| Im2 | Unclear needs | | 0.716 | | |
| Im3 | Lack of demand | | 0.587 | | |
| Im4 | Lack of knowledge about the technology | | 0.584 | | |
| Im5 | Low technical ability | | 0.573 | | |
| Im6 | Immaturity | | 0.536 | | |
| Im7 | Complexity | | 0.519 | | |
| | Privacy | | | | |
| P1 | Power availability | | | 0.863 | |
| P2 | Inadequate training | | | 0.651 | |
| P3 | Low investment | | | 0.610 | |
| P4 | Privacy | | | 0.496 | |
| | Security | | | | |
| S1 | Commitment from other project participants | | | | 0.721 |
| S2 | Time frame to develop the technology | | | | 0.710 |
| S3 | Inefficient use of software | | | | 0.566 |
| S4 | Security | | | | 0.554 |
| | % of variance | 32.321% | 30.817% | 18.677% | 18.185% |

### 4.2. Common Method Variance

Examination of a single factor was applied to evaluate the inconsistency of the standard approach [89]. If the total variance of the factor is less than 50%, the standard method variance (CMV) did not influence the data [77]. The findings also revealed that the original components explained 32% of the total variance. It implied that the CMV did not influence the results since it is less than 50% [77].

### 4.3. Measurement Model

The analytical assessment model includes an estimation of (i) composite reliability, (ii) indicator reliability, (iii) discriminant validity and (iv) average variance extracted (AVE), as argued by Hair, Jr. et al. [90]. This study used the PLS algorithm following Wong [82], weighting lessons, weighting scheme, path weighting, variance 1, data metric with mean 0, initial weights of 1.0, abort criterion of $1.0 \times 10^{-5}$ and maximum iterations of 300.

In general, indicators having external loadings varying from 0.4 to 0.7 must be measured for exclusion from the scale only if the indicator results removal led to a significant rise in AVE and composite reliability [91]. The variables having 0.65 as external loadings were categorized as uncompliant with this criterion and, as recommended, were excluded from further analysis [90]. It implied that the level at which roughly 50% of the indicator variance was described by its component and the stage at which the described discrepancy is above the error of variance. Additionally, Figure 3 and Table 3 show the variables' external loadings for the modified and original models.

**Table 3.** Construct reliability and validity tests.

| Constructs | Item | Outer Loading | | Cronbach's Alpha | Composite Reliability | AVE |
|---|---|---|---|---|---|---|
| | | Initial | Modified | | | |
| Infrastructure | In 1 | 0.67 | 0.662 | 0.802 | 0.858 | 0.504 |
| | In 2 | 0.735 | 0.767 | | | |
| | In 3 | 0.343 | deleted * | | | |
| | In 4 | 0.708 | 0.713 | | | |
| | In 5 | 0.531 | deleted * | | | |
| | In 6 | 0.708 | 0.744 | | | |
| | In 7 | 0.647 | 0.621 | | | |
| | In 8 | 0.728 | 0.74 | | | |
| Immaturity | Im 1 | 0.636 | 0.63 | 0.809 | 0.862 | 0.512 |
| | Im 2 | 0.449 | deleted * | | | |
| | Im 3 | 0.747 | 0.746 | | | |
| | Im 4 | 0.773 | 0.782 | | | |
| | Im 5 | 0.684 | 0.684 | | | |
| | Im 6 | 0.654 | 0.646 | | | |
| | Im 7 | 0.778 | 0.788 | | | |
| Privacy | P1 | 0.773 | 0.775 | 0.747 | 0.854 | 0.661 |
| | P2 | 0.825 | 0.822 | | | |
| | P3 | 0.839 | 0.84 | | | |
| Security | S1 | 0.788 | 0.79 | 0.751 | 0.843 | 0.573 |
| | S2 | 0.803 | 0.803 | | | |
| | S3 | 0.715 | 0.713 | | | |
| | S4 | 0.719 | 0.719 | | | |

* Deleted items.

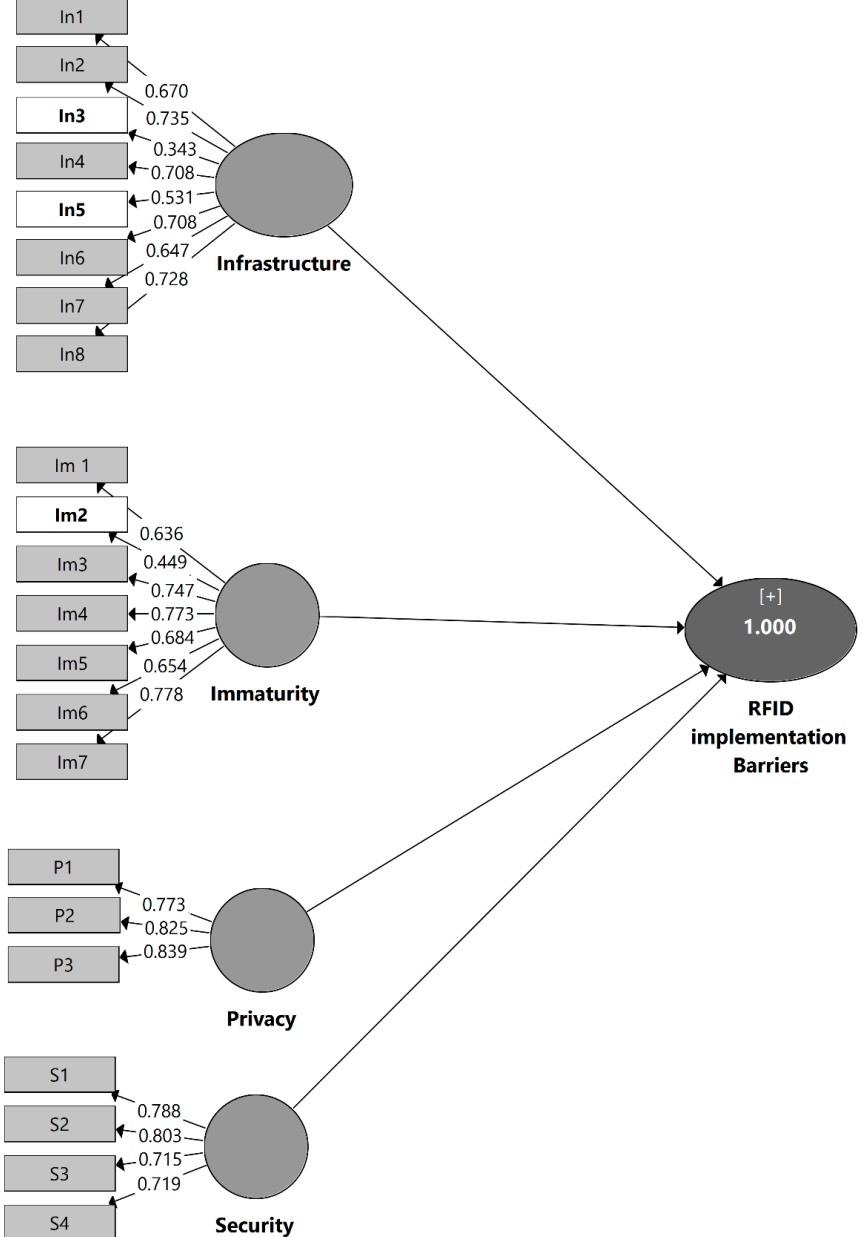

**Figure 3.** Structural model.

Therefore, all the external loadings from Im2, In3 and In5 were eliminated from the initial analytical model due to poor factor loadings below 0.5. It has substantiated their insignificant impact on the associated constructs. Further, a modified model was assessed after excluding the insignificant variables due to Cronbach's alpha limits. It estimates sensitivity toward the number of variables convoluted and the internal reliability of composite reliability (cr), which was also analyzed by Hair, Jr. et al. [90]. For this study, cr values higher than 0.7 were deemed acceptable, based on Hair, Jr. et al. [90].

Similarly, cr values higher than 0.6 are deemed appropriate for exploratory analysis [82]. The entire models attained a cr threshold value above 0.7, as revealed by Table 3, and were therefore accepted. A typical method for assessing the constructs' convergent validity within the model is AVE, which had values higher than 0.50. It implied a suitable convergent value which concurred with [82]. The findings, as contained in Table 3, revealed that all concepts of the model pass the test.

After achieving a significant difference between the constructs using the observed standard, it is possible to clearly define the discriminant validity (DV). Therefore, establish-

ing discriminant validity describes the singularities not adequately defined by other model constructs [92]. The DV can be estimated using three (3) approaches, (i) hetotrait–monotrait ration of correlations (HTMT), Fornell–Larcker's (1981) [80] criterion and cross-loading criterion. The AVE's square root of individual constructs was compared to the correlations of a particular concept with the remaining concepts to assess the DV. The AVE's square root has to be greater than the correlations between the concealed variables based on the Fornell and Larcker [80] criterion. This study's findings established the DV of the analytical model, as summarized in Table 4 [93].

**Table 4.** Correlation of latent variables and discriminant validity (Fornell–Larcker).

| Constructs | Immaturity | Infrastructure | Privacy | Security |
|---|---|---|---|---|
| Immaturity | 0.715 | - | - | - |
| Infrastructure | 0.622 | 0.71 | - | - |
| Privacy | 0.526 | 0.599 | 0.813 | - |
| Security | 0.459 | 0.648 | 0.474 | 0.757 |

Conversely, many researchers have overruled the Fornell and Larcker [80] criterion of standard DV. Subsequently, Henseler et al. [94] suggested a different procedure for estimating DV, i.e., HTMT. It is a novel procedure for estimating the DV of variance-based SEMs and measuring what could be a precise correlation among the binary constructs if the two variables are precisely estimated, i.e., if the two constructs are consistently measured. The HTMT model was likewise applied in this paper to evaluate the DV. Hair et al. [83] argued that the HTMT value must be lower than 0.85 and 0.90. It implied that the two variables were different. If the constructs of the model are theoretically comparable, the HTMT values must be below 0.90. Suppose the HTMT values are below 0.85, then the model's constructs should be hypothetically diverse. The HTMT values for the analyzed research concepts are presented in Table 5. Consequently, the constructs revealed adequate DV.

**Table 5.** HTMT values.

| Constructs | Immaturity | Infrastructure | Privacy | Security |
|---|---|---|---|---|
| Immaturity | | | | |
| Infrastructure | 0.744 | | | |
| Privacy | 0.641 | 0.742 | | |
| Security | 0.582 | 0.81 | 0.608 | |

The third approach, i.e., the cross-loading criterion, was applied in this research to estimate DV. The approach estimates that the indicators' loading for a particular concealed variable must be above the loading on the remaining variable by row. It implied that the indicators' loading (or items) for the variables have to be bigger than the loading of the remaining constructs. Table 6 reveals that the fundamental indicators' loading of the apportioned concealed construct was more significant than the cross-loading on the remaining variables by row. The findings further revealed a significant level of unidimensionality for individual constructs.

### 4.4. Structural Assessment Model

As soon as the RFID tools were outlined as a formative concept, the collinearity among the construct's formative construct objects could be explored further by evaluating the variable inflation factor (VIF). The findings also revealed that all the VIF values were below 3.5. It implied that these subdomains have individually underwritten the higher-order construct. Additionally, a bootstrapping technique was employed to prefigure the impact of the path coefficients [95]. The entirety of the paths were statistically significant at the 0.01 level, as indicated in Figure 4 [79].

**Table 6.** Cross loadings to test discriminant validity of indicators.

| Barriers | Immaturity | Infrastructure | Privacy | Security |
|---|---|---|---|---|
| Im 1 | 0.63 | 0.174 | 0.135 | 0.265 |
| Im3 | 0.746 | 0.586 | 0.441 | 0.178 |
| Im4 | 0.782 | 0.459 | 0.504 | 0.427 |
| Im5 | 0.684 | 0.399 | 0.42 | 0.501 |
| Im6 | 0.646 | 0.387 | 0.185 | 0.29 |
| Im7 | 0.788 | 0.563 | 0.448 | 0.296 |
| In1 | 0.417 | 0.662 | 0.276 | 0.43 |
| In2 | 0.515 | 0.767 | 0.523 | 0.569 |
| In4 | 0.384 | 0.713 | 0.479 | 0.46 |
| In6 | 0.408 | 0.744 | 0.414 | 0.386 |
| In7 | 0.39 | 0.621 | 0.361 | 0.368 |
| In8 | 0.516 | 0.74 | 0.467 | 0.517 |
| P1 | 0.274 | 0.384 | 0.775 | 0.236 |
| P2 | 0.492 | 0.404 | 0.822 | 0.439 |
| P3 | 0.479 | 0.634 | 0.84 | 0.444 |
| S1 | 0.301 | 0.544 | 0.436 | 0.79 |
| S2 | 0.239 | 0.345 | 0.336 | 0.803 |
| S3 | 0.464 | 0.573 | 0.305 | 0.713 |
| S4 | 0.355 | 0.458 | 0.351 | 0.719 |

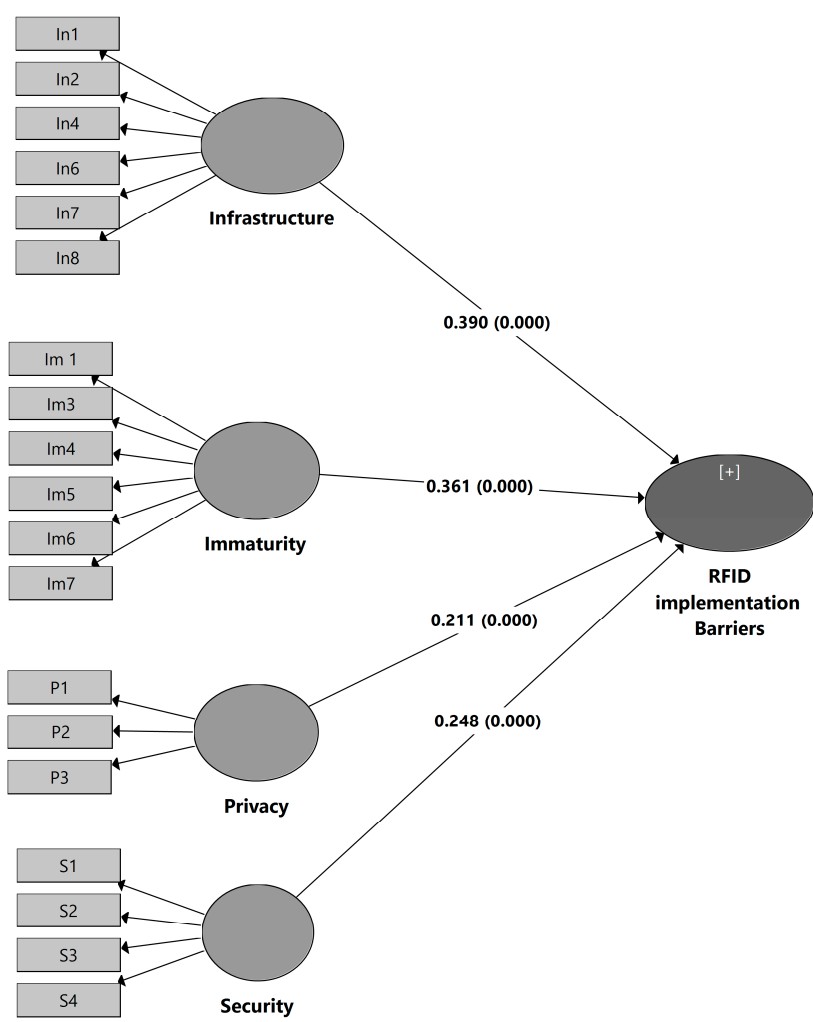

**Figure 4.** Performing bootstrapping analysis with path coefficients.

## 5. Discussion

Building schemes affect the economic and social aspects and the entire lifecycle of society [96]. Therefore, dynamic and viable processes, procedures and materials have been adopted [97]. Based on building experts' perspectives, it has been realistic that they encounter many questions if established and have a constructive commercial and social impact [98]. The building business involves significant viable development [99]. Therefore, RFID tools and procedures are significant for a perfect understanding of project execution. Hence, there is a dire need to identify the significant RFID barriers to facilitate its adoption. However, there are not many investigations on RFID tools' evaluations toward resolving the barriers, which could stimulate the adoption of RFID by building sectors in emerging countries. This research has attempted to narrow this gap by stressing the RFID hurdles to realizing the sustainability of building projects in Nigeria as a case study. The findings derived from EFA discovered that the RFID barriers might be classified into four groups founded on the analyzed data in this study. These barriers were grouped into immaturity barriers, security barriers, infrastructure barriers and privacy barriers. Detailed discussion on these barriers was attempted in the subsequent sections based on the recommended results of PLS-SEM model.

### 5.1. Infrastructure

The significance of 'infrastructure' is indisputable in RFID implementation. The PLS-SEM model suggested that this concept had the most analytical influence on RFID implementation hurdles. It had an outer coefficient of 0.39 through the 'Infrastructure Barriers' component. It contained six barriers: lack of maturity, low infrastructure, difficulty in changing the software, lack of industry standards, lack of communication in remote areas, and maintenance. These results have been agreed with by many scholars. According to Attaran [48], immaturity within the criteria was the major barrier to RFID implementation techniques. A significant portion of the studied participants (44%) believed that the current maturity in RFID criteria is adequate to deliver investment returns, though 29% agreed that the process manufacturers had attained an 'appropriate maturity level'. Likewise, 35% of large firms with over 5000 workers believed that standards are adequately matured to deliver investment returns [100]. In addition, one of the significant barriers to implementing RFID technology is inadequate infrastructure. Infrastructure that was not adequately put in place could make building project works ineffective [100]. Infrastructure included an excellent environment, power supply, network connections, etc. If these factors are not implemented, RFID technology will be challenging to implement [48]. However, due to the difficulty in operating, RFID technology is very complex; only a few professionals know how it works and can handle it [48]. Many professionals in the field would prefer the old method of monitoring site activities and tracking since RFID is complicated to operate.

Moreover, RFID technology is still confronted with challenges that could make full-fledged systems implementation by large manufacturers cost USD thirteen to twenty-three million [48]. The challenge concerning standardization between RFID constructors, tag frequencies, readers and filming software is also slowing RFID adoption due to low consumer confidence. In the same vein, the technologists are shrouding their data and evading standardization since it might offer a different builder the competitive advantage that transporters expect to achieve in making the RFID adoption the next major thing [101]. Based on the previous research, the following criteria could have an impact on the following issues: interoperability, data formats (encoding, syntax, data structure), international RFID frequencies, use of passive and active tags, presentation and identification methods and interactions between objects, tags and readers [102].

In terms of remote sensing, communication is an important aspect when it comes to handling building schemes. Communication serves as a means of passing information from one person to another or from one location to another. Lack of communication will slow down project works on site and affect the overall project performance and outcome. If there is a break in communication among professionals, RFID technology cannot be effective [53].

Furthermore, maintaining the RFID technology is among the significant challenges facing adopting RFID tools in the building business. Maintenance of RFID technology will help to reduce the cost of purchasing new wireless technologies, and it will help to extend the utilization of the machinery in the building activity [60]. However, RFID could be difficult to be maintained due to the professionals that are not familiar with it and do not know how it works. Additionally, money will be needed to maintain it, which might be an additional cost to a project, thus leading to the use of the manual approach [103].

### 5.2. Immaturity

Immaturity is one of the most critical barriers to RFID implementation in the building business. Implementing and using RFID technology is meaningless and a waste of time for some building industry people [104]. To some professionals in the building business, adopting and using RFID technology in tracking onsite activities is immature, and they prefer the traditional and old methods of going to the site and monitoring what is happening onsite [105]. The following principal component is connected to 'Lack of Maturity'. It had an outer loading of 0.361. This component comprised RFID barriers: virus threats, lack of demand, knowledge about the technology, low technical ability, immaturity and complexity. The results agreed with Huang et al. [55], who argued that all the data in the tags would be affected if there is a virus infection. Since it is a wireless technology and data can be stored in it for some time, it might be prone to various viruses. Thus, it is an obstacle to RFID adoption in the building business. Since not all professionals are familiar with RFID technology, the demand for the technology is correspondingly low. Only professionals familiar with the technology will use it and demand it; thus, the demand is reduced [53].

Similarly, RFID technology is not that common, especially in the building industry. Most professionals are ignorant of technology and will not succeed in its use in the field. According to Garfinkel et al. [106], no laws regulate the use of tags; thus, legislation will be needed to guarantee safety to the public. In the meantime, the initial adopters, including Tesco and Walmart, could aid in neutralizing anxieties by advertising a comparable proposal. RFID in the building industry is typically used to sustain the increasing demand for quick and effective materials management since building projects are becoming more unique and complex and need more comprehensive administration [53]. This is the act of using machines and equipment and methods that are not the most advanced. Lack of technical ability or low technical ability will negatively impact the implementation of RFID technology.

Finding appropriate and qualified professionals to carry out the works of the RFID technology is very difficult because not many are familiar with the technology that is being used [60]. Consequently, due to the difficulty in operating, RFID technology is very complex; only a few professionals know how it works and can handle it. Because of its complexity, many professionals in the field will prefer the old method of monitoring site activities and tracking since RFID is complicated to operate.

### 5.3. Security

The third principal component is related to "Security". Security is one of the significant difficulties circumscribing the confidentiality of data captured and the features related to the disposition of RFID [107]. In building schemes, employees entrenched with RFID must not be tracked after leaving the building location, observing their civil liberties. Conversely, there are some limitations for guaranteed tags and items that are utilized to track those items as transportation of goods. The challenges related to RFID security, otherwise identified as intrusion detection, denote the detection of external attacks upon the structure, usually using the tags that avert the total integrity of the data [108]. This involves barriers, such as commitment from other project participants, time frame to develop the technology, inefficient use of software and security. Security, with an outer coefficient of 0.248, ranked third on the RFID barriers scale.

In carrying out a project, the client is the overall manager and owner of the project. He finances it and decides what he wants. He employs other professionals to help him bring the project to reality. The professionals can determine the choice of technology to be used. In building, RFID is quickly changing the course of building processes. It can monitor and track activities onsite without requiring someone to be physically present [53]. Except for the professionals familiar with this technology, others might not necessarily commit to this technology used in most projects, thereby serving as a barrier to implementing the technology. In addition, to produce RFID technology, there must be enough time and patience to avoid producing fake ones. Shortage of time was the most critical hurdle to using RFID technology [103].

Compared to other tracking technologies, producing fake RFID technology is easy, but producing and using the accurate RFID technology will take some time to produce. According to Huang et al. [55], accurate RFID technology will have good storage capacity compared to the fake one, and thus, it takes time to produce. However, for professionals to enjoy using RFID, it must be used properly [100]. However, people in the field that are not that familiar with the use of the technology, or even its existence, is one of the barriers to implementing the RFID technology since it cannot be used efficiently and effectively.

*5.4. Privacy*

Within the context of an RFID, facilitated firm refers to inadvertently discharging critical data; the inherent problem is that radio-based tools inter-relate via invisible transmission frequencies; thus, it could be not known when communication is taking place [109], and all the variables currently discovered on a particular individual could not be accumulated [110]. The last component affecting the success of RFID adoption was 'Privacy'. It had an outer loading of 0.211. It comprised barriers including inadequate training, power availability, power supply, low investment or inadequate training that could impact the adoption of RFID. The RFID tools use power directly and indirectly because if the power of the tag is low, it goes off automatically.

The power supply is not constant, and since RFID technology makes use of power before it can operate, there must be a steady and good current power supply for the technology to work perfectly well and bring out the desired result [103]. Additionally, insufficient or inadequate training on the application of RFID tools will affect the implementation of the technology. Any professional unfamiliar with the technology's workings and functions will surely get it wrong when operating it [60]. Since there are not several experts in the building sector that can use the technology or even train others on how to use it, this is a barrier to implementing the tools in the building business [103]. In the meantime, investors and companies will not invest in a technology they are unfamiliar with due to the risk allocated to the technology investments that will bring little or no result. The low investment in this technology is among the major obstacles faced in executing RFID technology in the building industry. Professionals do not trust the technology, and they prefer to go for the manual and traditional approach of tracking and monitoring site activities, thus bringing about low investment in RFID technology [111].

## 6. Theoretical and Managerial Implications

Although the concept of developing a sustainable idea is not new [112], it is apparent in concert with a progressively essential function in a variety of businesses [113]. The recommended ranking technique is a significant hurdle to RFID implementation, especially in sustainable building. The recommended model was employed in this study to distinguish the RFID adoption barriers. Therefore, the gap between RFID theory and practice has been lessened in this study. However, based on the accessible literature, there were few studies concerning removing barriers to RFID adoption in Nigeria's building industry. Initially, the study methodically analyzed the significant RFID barriers that could aid RFID application in the building industry. This inference laid the foundation for upcoming research concerning the obstacles to RFID in emerging economies, particularly building management. The

conceptual components of this study offered a measurement context for measuring the RFID hurdles that could be successfully used in Nigeria and similar third-world nations.

The four components of RFID adoption barriers in Nigeria's building industry were analyzed using the novel PLS-SEM. Therefore, this research offered a framework for aiding stakeholders who are interns in integrating RFID objectively. Additionally, this research has significantly influenced the areas outlined below. These areas have momentous outcomes for the building industry:

- It offered a list of RFID adoption obstacles and the associated measurements to measure the significance and how those hurdles can be resolved to facilitate RFID adoption.
- It could aid constructors, consultants and clients examine and remove the RFID adoption barriers to enhance the planning and accuracy of building projects.
- It has demonstrated a methodical proof that could aid emerging countries in adopting RFID by resolving the prevailing barriers.
- The latitude of the RFID analyses has typically focused on advanced countries. Thus, there is limited research concerning RFID implementation barriers, especially within Nigeria's building domain.
- Consequently, this research effectively related RFID to Nigeria's building sector. It laid a robust basis for solving RFID implementation barriers by improving the sustainability of local building schemes and narrowing the existing research knowledge gap.
- It also offered a framework that can guide the policymakers concerned with the balanced development of RFIDs. This study established a novel prediction tool using PLS-SEM to examine RFID implementation barriers in the building industry in a third-world country.
- Therefore, this framework could be a game-changing tool in building schemes, especially in emerging economies. Besides the research conducted in Nigeria, it is expected that this model shift will stimulate analogous barriers and limitations in other emerging nations.
- The results will also facilitate the application of Nigeria's building schemes. Current results have provided a basis for knowledge goals concerning RFID adoption, including avoidable costs and apportioning reasonable budgets to individual projects.
- Therefore, by establishing and tracking the envisioned plans, major stakeholders might focus attention on the goal of the project regarding spending, productivity and time. Furthermore, devising a high level of sustainability in a project has a significant impact. Finally, building projects with a high sustainability standard will have positive benefits.

## 7. Conclusions

The literature was concordant vis-à-vis the poor quality of building schemes in developing nations. Notwithstanding the heavy dependence on RFID in several countries, its manifestation in third-world nations is relatively new. Like most emerging nations, Nigeria also experiences discrepancies and abnormalities in building quality and large-scale projects. This has flagged the need for RFID implementation to improve this situation. This study has revealed that RFID is a sustainable solution to curb that menace. Conversely, the adoption of this technique by the building industry in most notable emerging nations is still feeble. The proposed model in this study was logically established by the PLS-SEM technique using participants from Nigeria's building participants.

The results of the suggested model framework will aid building participants in resolving the barriers that will affect the adoption of RFID. Thus, it will reduce costs and increase sustainability in Nigeria and similar emerging economies. The findings could likewise aid constructors in realizing that RFID requires continuous training to offer customer satisfaction concerning building tasks and improves the customers' confidence in the business. This mechanism can help successful and efficient project management through RFID adoption. Thus, it will guarantee effective time management and cost reduction while increasing projects' quality. Therefore, all concerned stakeholders and organizations could

focus on the purpose of the projects concerning time, efficiency and cost by progressing the established policies. Finally, there is a good understanding concerning the need for achieving a high level of achievement in a project. These results would be advantageous in helping owners, professionals and developers assess and use RFID tools to facilitate consistency, planning and sustainability of building projects.

**Supplementary Materials:** The following supporting information can be downloaded at: https://www.mdpi.com/article/10.3390/su15010825/s1.

**Author Contributions:** Research Idea: A.F.K.; Conceptualization, A.F.K.; Methodology, A.F.K. and M.E.; Formal analysis, A.F.K.; Resources, A.F.K., F.S.K. and M.E.; Data curation, A.F.K., A.E.O., M.E. and F.S.K.; Writing—original draft, A.F.K.; Writing—review & editing, A.F.K., A.E.O., M.E. and M.M.H. All authors have read and agreed to the published version of the manuscript.

**Funding:** This study is supported via funding from Prince Sattam bin Abdulaziz University project number (PSAU/2023/R/1444).

**Institutional Review Board Statement:** Not applicable.

**Informed Consent Statement:** Not applicable.

**Data Availability Statement:** Not applicable.

**Conflicts of Interest:** The authors declare no conflict of interest.

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
