# Peer review of "Barriers to the Implementation of Radio Frequency Identification (RFID) for Sustainable Building in a Developing Economy"

_sustainability, doi:10.3390/su15010825_

Round 1

Reviewer 1 Report

The main contribution of the paper is the barrier identification for RFID to be deployed in the construction industry of developing countries.  The structure of the paper sets out introduction of different components and methods in Secs. II and III.  These two sections need to be significantly condensed to make the paper length tractable.

The major contribution is unclear.  Why is the survey finding representative or useful for further RFID adoption?  The statistical method seems to be robust but needs to be presented in a more concise manner.

Unclear writing (maybe typos that show the poor paper quality):

* What is "projects night" in line 90?

* What is FRID in lines 98 and 102?

* KOM or KMO in lines 339 - 346?

* None of the auxiliary info is provided in lines 679 - 697.

* Reference format is not consistent.  Several cited paper titles capitalize every word, but most follow IEEE format.  Even as a survey article, the number of references should be condensed.

Reviewer 2 Report

1) Show the table 1 in the form of grids so as to make it easily readable to all the readers

2) Why didn't the authors calculate the rms value in Table 2 ? Why did they calculate %age variance ?

3) Section 5 needs to be shortened. Some of the matter is redundant in this section.

4) Please mention the exact finding from the literature review. 

5) Please mention whether you are presenting your own model or modifying the existing model 

6) There are some improvements needed in the flow of the paper

7) Read and refer the following papers in your article. These might be very useful for you.

Gupta, A., Asad, A., Meena, L., & Anand, R. (2023). IoT and RFID-Based Smart Card System Integrated with Health Care, Electricity, QR and Banking Sectors. In Artificial Intelligence on Medical Data (pp. 253-265). Springer, Singapore. 

 Srivastava, A., Gupta, A., & Anand, R. (2021). Optimized smart system for transportation using RFID technology. Mathematics in Engineering, Science & Aerospace (MESA)12(4).

Reviewer 3 Report

The article investigated a very interesting topic about identifying the barriers of RFID adoption in the construction industry. The authors determine RFID barriers and their significance based on surveying various professionals in the construction industry in Nigeria. I would like to provide the following comments and insights to improve the work:

1. The two terms "construction" and "building" have been used interchangeably, although "construction" is more generic. I would suggest using one term throughout the article for better consistency and to reduce confusion.

2. The purpose and objective of the study are not clear in the introduction section. Although the authors mentioned the aim of the study in 98-102, the term "cloud computing" might be confusing in the sense that it was not clearly linked to the RFID which is the main focus of the article.

3. The self-citing in the second sentence in the introduction section seems unnecessary (lines 38-40). The information is common and it does not seem to convey the purpose of the previous study.

4. In general, the English language requires improvement and needs professional proofreading. Some sentences are not complete or grammatically missing verbs or nouns. For example, the sentences in lines 51-52 and 605-606 are incomplete.

5. In the introduction section, the description of the RFID is unnecessarily repeated in lines 63-68 and 77-81.

6. There are a few spelling mistakes and some acronyms are not defined or are typed wrongly. For example, "FRID" in lines 98 and 102, and "RFIDD" in line 135. Also, "UHF" in line 219 and "AMR" in line 501 are not defined. Moreover, sometimes you use "PLS-SEM" and sometimes you used "SEM-PLS".

7. Format of references is not consistent. For example, the link [3] is not formatted correctly. Also, in [35] the DOI is not placed correctly.

8. In Table 1, "Lack of Maturity" and "Immaturity" seems the same. These two barriers can be combined instead of being listed as two distinct lines.

9. Studies described in lines 231-237 need proper referencing.

10. Section headings are repeated "4. Results and Discussion" and "5. Discussion".

11. What are some the questions used in the questionnaire? It would be more appropriate to provide some samples and explain the logic behind them.

12. In many occasions, the authors mentioned that the study provide a methodological approach to resolve the barriers. However, this is not clear anywhere in the manuscript as the main focus was to identify the barriers and their significance. The study does not clearly focus on providing solutions to eliminate those barriers.

13. Also, how the study objectives are linked to "sustainable" construction is not very clear. Very little literature and linkage to sustainable construction is present in the study. I would suggest providing a section or some write-up on how eliminating RFID barriers would support sustainable construction.

14. In Figure 1, you indicate that you used EFA while in line 258 under section 3.1, you mentioned that CFA was employed. I suggest showing CFA in Figure 1 as well if it was part of your methods.

15. In line 400, the authors refer to "cr values" in Table 5. However, the title of Table 5 is HTMT values which raise a concern about consistency and clarity of explanations. Are those values refer to the same measure? If yes, you should be consistent and use only one acronym.

16. The organization of the manuscript needs revisiting. It would be more useful and appropriate to explain more about the actual RFID barriers before an extended discussion about the different tests used. Also, an extended discussion on how other scholars had identified RFID barriers would be very helpful. You might refer to Valero et al. 2015 (doi.org/10.3390/s150715988) and Mohsen et al. 2022 (doi.org/10.1016/j.aei.2022.101631).

17. The explanation about the different tests and measures used are not well organized and it is not easy to follow what and why the authors are using those different tests. Also, how the test are linked to each other seems arbitrary. There are also some redundancy of using many tests to achieve the same conclusion. For example, using three approaches for DV: HTMT, Fornell-Larcker, and Cross-Loading. I suggest using only one approach and explain it very well.

18. In lines 669-670, "this study has provided a measure for the project team under Nigeria’s building schemes." This sentence is not clear. What "measure" you are referring to? What is being measured here?

Round 2

Reviewer 3 Report

 The article has been enhanced as compared to the first submittal. However, I would like to provide the following comments:

1. The revised version should have highlighted ONLY the sentences that were added or changed. Highlighting a whole section is not a good practice. Also, you need to specify the page and line number where the revision has been applied.

2. There are still some errors when using acronyms. In the abstract, check line 24.

3. What is "poly-makes" in line 95?

4. In line 97, which results do "these results" refer to?

5. In line 115, the sentence is grammatically missing a noun "Therefore, is a dire need ..."

6. The sentence in lines 119-121 is not clear. It can be simplified; no need to complicate it.

7. In line 204, it is "executed" not "execute."

8. In line 271, you mentioned, "The statistical analysis conducted in this paper included dimension and structural assessment techniques." Then, you listed several measures under two headings, 3.2.1 and 3.2.2; which is the dimension, and which is structural? It would be better to use consistent terminology and be clear about categorizing the techniques used in your study.
